# Prevalence and Demographic Distribution of Hypermobility in a Random Group of School-Aged Children in Nigeria

**DOI:** 10.3390/healthcare11081092

**Published:** 2023-04-11

**Authors:** Oluwakemi Adebukola Ituen, Ebuka Miracle Anieto, Gillian Ferguson, Jacques Duysens, Bouwien Smits-Engelsman

**Affiliations:** 1Department of Health & Rehabilitation, University of Cape Town, Cape Town 7701, South Africa; 2Department of Physiotherapy and Paramedicine, School of Health and Life Sciences, Glasgow Caledonian University, Glasgow G4 0BA, UK; 3Motor Control Laboratory, Movement Control and Neuroplasticity Research Group, KU Leuven, 3001 Leuven, Belgium

**Keywords:** joint hypermobility, range of motion, Beighton score, prevalence, school-aged children, localized hypermobility, generalized joint hypermobility, hypermobility spectrum disorder

## Abstract

Background: The purpose of this study was to determine the prevalence of hypermobility in randomly selected healthy children, without previous trauma or disease process affecting the joints and whether other demographic variables (age, sex, BMI) had an impact on Beighton scores and range of motion (RoM) in children between 6 and 10 years of age. Results: 286 children were included; 27.3% of them had a Beighton score ≥7/9 and 72% would be classified hypermobile if we had used a Beighton cut-off score ≥4/9. Prevalence declined with increasing age. Girls were more often hypermobile (34%) than boys (20%) and this was mainly caused by increased RoM in the knees. Positive scores of finger items of the Beighton were more common than on the other items, leading to a high prevalence of peripheral hypermobility. Localized hypermobility was only found in the fifth MCP joint. A total of 15% of the children with normal mobility reached 20 excess degrees RoM of the left and right fifth MCP. Pain was present in 12 of the 239 children but was not linked to the level of mobility. Conclusion: Hypermobility is the rule in this pain-free population of children with GJH.

## 1. Background

Joint hypermobility is a condition characterized by excessive passive and/or active range of motion of the joints beyond normal limits along physiological axes [1]. This excess movement could be peripheral (limited to the hands and feet), localized (involving a single joint), or generalized (involving multiple joints), referred to as Peripheral, Localized, and Generalized Joint Hypermobility (GJH), respectively [2,3]. Joint hypermobility is predominately genetic in nature but can also be a consequence of trauma, malnutrition, or exercise [4,5]. Several tools have been used to identify hypermobility, but the most frequently used tool has been the Beighton scoring system [6].

Unlike GJH, localized and peripheral joint hypermobility have been scarcely reported in the literature. The prevalence of GJH varies between 8.8% and 64.6% taking age, gender, ethnicity, and Beighton score cut-off into consideration [7,8,9,10]. According to Castori and Hakim, generalized hypermobility may be observed in the joints of 2–34% of males and 6–57% of females [10]. De Boer et al. used a Beighton score cut-off of >4, >5, or >6 to identify GJH in a population of 245 Dutch children and reported varying prevalences of 34.1%, 22.5%, and 16.5%, respectively [11]. Furthermore, a study conducted among 303 Arab school-aged children identified 15.2% and 7.6% of the study population as hypermobile using a Beighton cut-off of ≥4 and ≥6, respectively [12].

Two university-based studies carried out in Turkey and the United States of America reported the prevalence of localized joint hypermobility in their studies as 21.5% and 57.5%, respectively [13,14]. In addition, the Turkish study reported the prevalence of peripheral hypermobility in their study as 4.2% [13]. While it is believed that peripheral hypermobility is more common in children, we are not aware of any study on localized or peripheral hypermobility in children.

The influence of ethnicity, sex, and age on the prevalence of GJH has been described in two narrative literature reviews [15,16]. According to these reviews, GJH is most prevalent in infants but decreases fast during childhood and slower during adulthood; Caucasians have shown a trend of a lower prevalence of GJH than Africans, Asians, and Arabians, and females have a higher prevalence of GJH as opposed to males [15,16]. Interestingly, some studies have found no gender difference in the prevalence of GJH [3,9].

The Beighton scoring system was modified by Beighton and Horan and has been used to identify GJH [17]. The Beighton scoring system has been most frequently used in epidemiological studies because it is easy to perform [10,18]. However, its use as a diagnostic tool is under consideration as it lacks a consensus-based cut-off [16,19]. The British Society of Rheumatology recommended a cut-off of ≥4 and this has been commonly used to define GJH followed by a cut-off score of ≥5 [19]. In their study of 1845 Swedish children, Jansson et al. observed that a Beighton score cut-off of ≥4 for all ages and gender will result in an overrepresentation of GJH, especially among young children and females [20]. In a recent study by Singh et. al., they found a 60% false positive rate of GJH through a logistic regression analysis supporting the fact that there is a high likelihood of misclassifying young children and females as hypermobile when a Beighton score cut-off of ≥4 is used to identify GJH [19]. Recently, the International Consortium on Ehlers–Danlos Syndrome reached a consensus on age-based Beighton score cut-offs with ≥6 for pre-pubertal children and adolescents, ≥5 for pubertal males and females up to the age of 50, and ≥4 for those >50 years of age [21].

Furthermore, the dichotomous pattern of the Beighton scoring system only shows the presence or absence of hypermobility and not the severity of the excess movement at the joint [3,19]. Although not all hypermobile joints are unstable, previous authors have linked laxity of ligaments and joint instability, to the subsequent onset of musculoskeletal symptoms such as pain, dislocation, or neurodevelopmental problems such as poor motor coordination and motor delay in GJH [2,10]. Considering the fact that there are various degrees of excess movement in individuals with GJH, the question remains: What amount of excess movement is a precursor to the development of musculoskeletal symptoms [6,20]? That is, will individuals with higher degrees (e.g., 20 degrees) of hyperextension show a higher risk of developing symptoms compared to those with lower degrees (e.g., 10 degrees) of hyperextension? Jelsma et al. argued that the inclusion of goniometric measurement of the joints to the Beighton scoring system will give an objective identification of hypermobility and a possible explanation for the development of musculoskeletal symptoms later in life [22].

Joint hypermobility provides flexibility that facilitates excellent motor performance as demonstrated by gymnasts and ballerinas. A study of elite sportsmen by Schmidt et al. demonstrated a higher prevalence of GJH among ballerinas (68.2%), than handball players (13.2%) [23]. Despite this benefit, approximately 3.3% of women and 0.6% of men with GJH develop musculoskeletal symptoms later in life [24]. When GJH becomes associated with musculoskeletal symptoms it is referred to as Hypermobility Spectrum Disorder (HSD) [2]. The diagnosis of HSD has been frequently missed or delayed due to a paucity of information regarding its etiology [25,26]. Therefore, an effective method of identifying children with GJH and those who are susceptible to developing musculoskeletal symptoms is needed so that interventional approaches can be tailored early, thereby preventing complications [27].

Even though the literature has paid attention to GJH over the years, less is known about Local Joint Hypermobility (LJH). Two studies have reported a higher prevalence of LJH than GJH in adult populations; also, one of the studies reported a higher frequency of injury among the participants with LJH [14,28]. What is still not known is if the prevalence of LJH and its association with musculoskeletal symptoms will show the same trend in children.

Although it has been established in the literature that, in general, females are more mobile than males, evidence of gender differences on the different Beighton items has not been conclusive in previous studies [9,12,20,29]. The inclusion of hands-on-floor as a joint hypermobility test has been debated, as it is highly influenced by training [4]. In addition, the trunk, pelvis, and lower limbs are considered a dynamic structure with associated functions, such that tightness of the hamstrings can limit trunk flexibility [30,31]. A study involving 460 school-aged children reported that 86.16% of children with GJH (Beighton cut-off of ≥5) could not perform the hands-on-floor maneuver [8]. The study further concluded that the hands-on-floor maneuver has a very low sensitivity (13.84%) in identifying GJH and that it does not have an additional value in the Beighton score [8].

Although the Beighton scoring system was first used to evaluate the prevalence of GJH in a South African population of children [17], there is a paucity of data on the prevalence of GJH among Nigerian children. Therefore, this study aimed at exploring the following research questions:What is the prevalence of PJH, LJH, GJH, and HSD among Nigerian children, and is this prevalence different for age, sex, and body mass index (BMI) classification groups, and how does it relate to prevalence in other countries?Which specific items of the Beighton scoring system are the most frequent in contributing to the classification of hypermobility and which ones are the least?What is the degree of excess movement in the fifth fingers, elbows, and knees in children classified as normal RoM, mobile and hypermobile?

The inclusion of goniometric RoM of the joints in this study provides additional value to the Beighton criteria in identifying joint hypermobility. This may help identify individuals within the upper limit of joint hypermobility, thus improving both the sensitivity and specificity of the Beighton scoring system. In addition, it may also provide insight into whether the various excess degrees of movement increase the risk of developing musculoskeletal symptoms in a group of individuals with GJH.

## 2. Methods

### Design

The study used a cross-sectional analytical design.

#### Participant Description

Study participants were recruited from 11 schools in the Southern part of Nigeria through convenience sampling. The study included children in grades 1–4 between the ages of 6–11 years. All parents of the children from the selected classes were given information and consent forms. Only children whose parents gave informed consent and children who gave assent were included in the study. Any child with a recent musculoskeletal injury, or physical disability as reported by a parent or medical doctor was excluded from the study (see flow chart in Figure 1). Children who reported febrile illness on the day of assessment were tested after recovery. To check for eligibility, the parents gave background information on the child’s health status and completed the child’s physical activity readiness questionnaire (PARQ) [32]. A total of 445 children were potentially eligible for inclusion, however, the parents of 159 children did not provide consent in time. Therefore, these children were excluded from the study leaving a total of 286 children who participated in the study (see flow chart in Figure 1).

## 3. Measurements

### 3.1. Anthropometric Measures

We collected data on participants’ age (years), sex, height (centimeters), and weight (kilograms). Height and weight were measured using measuring tape and a weighing scale (on bare feet; measured to the closest one cm and 100 g, respectively). The body mass index (BMI) calculation was performed using the age-gender-specific BMI centiles recommended by the WHO [33]. The BMI centile classifications are: Underweight = ≤2nd centile, Normal weight = 3rd to 90th centiles, Overweight = 91st to 97th centiles, Obese = ≥98 centile.

### 3.2. Joint Mobility

Joint hypermobility was classified based on the scoring system developed by Beighton et al. [17]. The test consists of four passive range of motion (RoM) items (assessed bilaterally) and one active forward flexion task (Table 1). The Beighton scale has a 9-point maximum score based on the degree of movement of the 5th metacarpophalangeal (MCP), elbow, knee joints, thumb movement, and the mobility of the spine. Participants were awarded one point for a positive test, and a total numerical score of 0 to 9. Although the Beighton test has not been formally evaluated for its psychometric properties in the Nigerian population, it is considered a valid test of joint mobility in children [3], which was shown by comparing the Beighton scale to the goniometry of 16 passive ranges of motion of joints on both sides of the body. Joint mobility was categorized based on the total Beighton score obtained. The 9-point Beighton score can be divided into 3 main categories (category I Normo-mobile = 0 to 4 points; category II Mobile = 5 to 6 points; category III Hypermobile = 7 to 9 points) with the highest category representing participants with the greatest joint hypermobility. Normo-mobile, Mobile, and Hypermobile categories represent participants who had normal joint mobility, moderate hypermobility, and generalized joint hypermobility, respectively.

### 3.3. Goniometry

Additionally, the joint range of motion of the 5th MCP joint extension, elbow joint hyperextension, and knee joint extension was assessed to the nearest 1 degree using the standardized joint mobility protocol [3]. A standard 2-legged 360 degree type Collehon extendable goniometer (Lafayette Instrument Company, Lafayette, IN, USA) was used for the knees and elbow, and a small arm goniometer was used for the 5th MCP. Joints were measured bilaterally. 

### 3.4. Pain Scale

To examine pain, participants were asked to choose the face from the Wong–Baker Faces Pain Scale (FPS) that best reflected the intensity of the pain they were experiencing [34]. The FPS (Figure 2) is a self-report pain scale that uses facial expressions to assess the intensity of pain. It is a valid instrument and has been used to assess pain among Nigerian children with sickle cell anemia [35]. 

## 4. Data Analysis

Data were analyzed using SPSS version 28. Descriptive statistics (frequency, percentage, mean, and standard deviation) were used to present the demographic data, Beighton scores, RoM, and Beighton classification of the study population. The univariate ANOVA was used to test for differences between the three joint mobility groups, in age and BMI classification. Because of the lower numbers of participants in the 10- and 11-year-old bracket, these data were combined for the age analysis. The children in the overweight and obese categories were also combined because of their low numbers.

To evaluate differences in the frequency of Beighton items and RoM between age groups, sex, and BMI classification groups, standard tests of significance were applied. Pearson Chi-square tests were used to test if the prevalence of positive and negative Beighton items was different for age group, sex, and BMI classification.

For the goniometry, paired t-test was used for the comparison between RoM for the left and right sides and an independent t-test for gender differences. Lastly, ANOVA was used to test if RoM was different for age and BMI classification groups. Degrees greater than the Beighton criteria, extension for elbow and knee (≥10), and 5th MCP (≥90), were designated as “excess degrees of motion”. Alpha was set at 0.05 for a 95% confidence level.

## 5. Results

### 5.1. Participants’ Data

The study included a total of 286 children, 138 males (48.3%) and 148 females (51.7%). Mean age 7.7 (SD 1.2). Of the children 99.3% were right-handed. Mean BMI was 15.34 (SD 3.0). Almost half the children (49.3%) were classified within the normal range of weight for their age and sex, and 38.8% were underweight. Overweight and obese accounted for 8.4 and 3.5%, respectively.

### 5.2. Prevalence of GJH

The mean Beighton score was 5.35 (SD 1.77). Joint mobility, classified based on the total Beighton score, yielded 35.3% normo-mobile children (score 0–4), 37.4% mobile children (score 5–6), and 27.3% hypermobile children (score 7–9). Demographic data per joint mobility group are shown in Table 2. 

If we examine the frequency of the dichotomous classification (negative/normal range or positive/hypermobile) per item, the 5th MCP joints had the highest frequency of positive scores in both males and females (98.3%; see Figure 3).

Goniometry showed that the overall excess range of motion was larger on the left side than on the right side of the body (t(1285) = 4.52, *p* < 0.001). The means of the passive RoM of the 5th MCP (right = 7.40 degrees (SD 7.96), left = 8.42 degrees (SD 7.88); t(1285) = 5.67, *p* < 0.001), elbow (right = 0.44 degrees (SD 4.12), left = 0.91 degrees (SD 4.12); t(1285) = 2.90, *p* = 0.004), were higher on the left side and not different for the knees (right = −2.11 degrees, left = −2.07 degrees; *p* = 0.77). 

### 5.3. Differences in Beighton Classification Per Age, Sex, and BMI Classification

Details of the age, sex, and BMI differences on the Beighton classification are presented in Table 2. 

Age: Hypermobility was less frequent in older children (Χ² = 27.34, *p* = 0.007; see Figure 4). The mean Beighton score decreased with age (F(4281) = 7.77, *p* < 0.001). The frequency of positive scores over age groups was not different for the item hands-on-floor. 

Based on the measured range of motion, there was a significant effect of age for all joints except for the left knee (*p* = 0.14; see Figure 5). However, some fluctuations were seen between adjacent age groups for the MCP5 joints.

Sex: The mean total Beighton score was 5.04 (SD1.67) and 5.64 (SD1.82) for boys and girls, respectively (t(1284) = −2.94, *p* = 0.004). Of the girls in our sample, 34% were classified hypermobile (score 7–9) while this was 20% for the boys. Females were more often rated hypermobile on the different Beighton items and the chi-square showed a higher frequency of positive scores for the left elbow (Χ² = 5.98. *p* = 0.015), and left (Χ² = 11.22, *p* = 0.001) and right knee (Χ² = 7.91, *p* = 0.006). One item showed a trend in the opposite direction; boys were more often (16.7%) able to put their hands flat on the ground than the girls (9.5%; Χ² = 3.29. *p* = 0.08). 

Based on the measured degrees of RoM, there was a significant effect of gender for the knees (right (t(1284) = −2.06, *p* = 0.04); left (t(1284) = −3.26, *p* = 0.001) and not for the MCP5 and elbows.

BMI: Differences emerged in Beighton scores between BMI classification groups, but the pattern of differences was not comparable between joints. Frequencies of positive scores were different for the knee (Χ² = 8.54, *p* = 0.014) and thumb item on the right (Χ² = 8.68, *p* = 0.013).

Goniometry showed that all except one joint were different (elbow right; *p* = 0.21) for the BMI classification groups (all *p* < 0.02), however, the direction differed per joint (Figure 6). The little fingers showed a larger RoM in obese/overweight children, while the knees were more mobile in underweight children. The elbows were the least mobile in the normal-weight children.

### 5.4. Prevalence of Localized Hyper/Hypomobility

Peripheral hypermobility was defined as at least 3 positive items in the hands of normo-mobile children and occurred in 12.9%. None of the normo-mobile children scored positive on all 4 hand items.

Localized hypermobility was defined as 10 degrees more RoM than a positive Beighton score in normo-mobile children. Out of 101 children with normal mobility as defined by the total Beighton score, 15 (14.9%) could move further than a 100 degree angle in the left and right 5th MCP joint. No other joint had an excess RoM in children with normal mobility.

Major joint hypermobility was defined as 10 degrees more RoM than a positive Beighton score in at least 3 joints (elbows and knees) was absent in the normo-mobile group. Out of the 185 children classified as hypermobile, 3 (3.6%) had 20 degrees RoM hyperextension in the right elbow joint and 5 (6.4%) in the left elbow joint. One mobile child had a hyperextension of more than 20 degrees in the left knee, and none of the mobile and hypermobile children had 20 degrees or more of hyperextension in the knee joints. 

Excess RoM was defined as degrees greater than the Beighton criteria, which, for extension of elbows and knees, is more than 10 degrees and, for 5th MCPs, more than 90 degrees. The excess degrees of motion in children classified as normal RoM, mobile and hypermobile are summarized in Table 3 and Figure 7. Excess range of motion is significantly different between mobility groups for all joints, as are post hoc comparisons (all *p* < 0.001) except for the comparison between the mobile and hypermobile groups for the MCP5 joints. As shown in Figure 7, there is a large overlap in Beighton classification groups for children with the same excess range of motion. Some children in the mobile group have a large excess range of motion but it is located in only a few joints or, in the case of normal mobile children, in even fewer joints.

Hypomobility: All children reached full extension in the knee and elbow joints (at least 0 degrees). Five (1.7%) children did not reach 90 degrees extension in the right 5th MCP and 6 (2.1%) in the left 5th MCP. One child had low flexibility in all joints measured.

### 5.5. Prevalence of Pain

Responses to the pain questionnaires were available for 239 children. A total of 94.1% of the children reported no pain, 8 children marked the face indicating “hurts a little bit” (3.3%; 3 normal, 2 mobile, 3 hypermobile), 6 children rated their pain as “hurts even more” (2.5%), none of them were hypermobile, 4 normo-mobile, 2 mobile.

## 6. Discussion

The first aim of the study was to describe the prevalence of Generalized Joint Hypermobility, Peripheral, Localized, and Hypermobility Spectrum Disorder among Nigerian children. As there is no universal agreement on a threshold for hypermobility, we used three cut-off scores to define joint mobility classifications: Normal Range (score 0–4), Mobile Range (score 5–6), and Hypermobile Range (score 7–9). Using this classification, 35.3% of the children were classified as Normal-mobile, 37.4% as Mobile, and 27.3% as Hypermobile. If we use a total Beighton score of ≥5 as a cut-off for hypermobility, a prevalence of 64.7% would be established. The higher prevalence seen in this study compared to previous studies may be due to two reasons: first, ethnicity and second, the age range [36,37,38].

The high frequency of GJH in our present study is supported by a study of an Australian population that included Caucasians and non-Caucasians (Africans, Asians, and Indigenous Australians). Hypermobility was significantly lower in the Caucasian group [19]. Interestingly, Morris et al. carried out a study in an Australian population and reported a prevalence rate of GJH that was 28.8% higher than a similar group in the United Kingdom [39]. Reuter and Fichthorn in their study of 654 American University students did not report a significant difference in the prevalence of GJH among the different ethnic groups and concluded that the influence of ethnicity needs future research [14].

In this study, we also presented the prevalence of Peripheral (17%) and Localized hypermobility (15%). Localized hypermobility was very rare, and only occurred in the finger joints. It is interesting to note that 15 children, though classified as normo-mobile, had an excess ROM of 20 degrees of the fifth MCP (right and left), while only three children classified as hypermobile had an excess of 20 degrees ROM of the fifth MCP (right and left).

### 6.1. Beighton at Item Level

The analysis, on an item level, of the Beighton scoring system showed that the fifth MCP joints had the highest frequency of hypermobility (98.3%) and largely contributed to percentages of children classified as hypermobile. This finding is consistent with previous studies where the frequency of hypermobility is higher in the upper limbs than in the lower limbs [12,39,40]. If only the larger joints (Major Joint Hypermobility: knee and elbow) were used to classify hypermobility in our study, the percentage would go down to about 39%. The hands-on-floor item has consistently shown a trend of least positive scores in previous studies in children, and our finding was not different, neither was it age-related [6,40]. The atypical finding in our study was the higher frequency of boys having a positive score compared to girls on this item. Typically, children in our age bracket find it difficult to place their palms on the floor because they are at the stage when the growth rate of their lower limbs is higher than the trunk and upper limb lengths [8]. Thus, the hands-on-floor item might give different results in adulthood. Interestingly, in an earlier study among a Nigerian population (a setting of peasant farmers and traders) aged 6–66 years, hands-on-floor had the highest frequency of hypermobility on the Beighton items [36]. Their study had a large age range but did not report the age at which the trunk flexibility started to increase. The habitual posture of bending over is common among farmers and may have helped to stretch the hamstrings over time. Previous authors have questioned the inclusion of “hands-on-floor” in the Beighton items as this test is not solely based on the laxity of ligaments but also hamstring extension and the individual’s growth stage [8]. In addition, the dynamic relationship between the thorax, trunk, and lower limbs makes it impossible to treat their functions in isolation [37]. Interestingly, the study by Czaprowski et al. reported no difference in pelvic-hip muscles and trunk flexibility in children with and without GJH [31]. A consideration for future studies is to evaluate the suggested association between lumbopelvic control and risk of injury in the lower limbs, as this may provide justification for the inclusion of hands-on-floor in the Beighton test [31]. 

### 6.2. Goniometry

Because the Beighton scale gives no indication of the degree of hypermobility we also used goniometry to measure the RoM of the MCP5, elbow, and knees, as we have shown this to be a valid way of measuring hypermobility [3]. By doing so, we could show a large overlap in the additional range of motion in the six joints measured in the three mobility classification groups. In theory, a child classified as Normal-mobile with two mobile joints (e.g., two times 15 degrees hyperextension knees) could have more excess RoM than another child classified as Hypermobile with 6 times 1 degree additional hyperextension in the knees, elbows, and little fingers above the Beighton criterium. Especially for follow-up studies, goniometry may be more sensitive than a dichotomous scaled item and might show higher relation with the development of musculoskeletal complaints. Malek et al. argued that localized hypermobility may even have a higher risk of becoming symptomatic than GJH [6]. In the present study, we also showed that the overall excess range of motion was significantly larger on the left side than on the right side of the body. Our sample population was predominantly right-handed making the left side non-dominant. This is in line with our previous study on Dutch children [3].

### 6.3. Age

The inclusion of 6 years old children will have increased the frequency of hypermobility, which corroborates with earlier findings [6,12,19]. Our study found a significant decrease in the mean Beighton score with age. The hypermobile group had the lowest mean age (7.29 years) while the normal mobile group had the highest mean age (8.03 years). When goniometry outcomes were compared, we found a significant effect of age for all the joints except for the left knee. 

### 6.4. Sex 

In this study, the prevalence of GJH was significantly higher among females. This is in line with previous evidence in the literature [41,42]. Based on the dichotomous classification of the items, females had a significantly higher frequency of positive scores for the left elbow (*p* = 0.015), right knee (*p* = 0.006), and left knee (*p* = 0.001). If we used goniometry, gender differences were only found for the knee joints. This is an important finding as it might be related to the development of symptoms later in life [42]. Especially, in combination with increased valgus angles more often seen in females than in males [43]. 

Interestingly, in the hands-on-floor test, males had a higher frequency of positive scores in this test, although the difference was not significant (*p* = 0.08). Even though the literature has widely supported a higher frequency of GJH among females than males due to hormonal influence, the evidence has been inconclusive and may again depend on age and cut-off values used. Studies by Smits-Engelsman et al. and Juul-Kristensen et al. included children between ages 6–12 years and both studies did not find an association between GJH and gender [3,38]. The studies by Singh et al. and Morris et al. included adolescents, and both reported a significant association between gender and GHJ [19,39]. In the study by Sirajudeen et al., the prevalence of GJH was not significantly different between genders; however, females were more hypermobile in their thumb and elbow, the males were more hypermobile in their knee and trunk [12]. 

### 6.5. BMI

In most developing nations, obesity is rapidly becoming a serious health issue but, in our sample, a third of the children were underweight. We examined weight as a risk factor for increased joint movement and the likelihood of (later) complaints. The present study found that the hypermobile group had the highest number of underweight children ≤2nd centile). We also found a significant difference between positive scores for hypermobility and BMI groups for the knees and right thumb items. Interestingly, goniometry showed an opposite trend in the range of motion and BMI classification for the fifth MCP joint and the knee; little fingers were more mobile while the knees were the least mobile in overweight/obese children. Although BMI is assumed to have an influence on GJH, the findings in the literature on their association have not been conclusive. In the study by Clinch et al. on 6022 children, the relationship between BMI and GJH was found among obese girls who were 2.47 times more likely to be classified as hypermobile [40]. In the study by Sirajudeen et al., the association between BMI and GJH was found among females with lesser BMI scores [12]. A recent study showed the impact of undernutrition on muscular power and agility [44]. The combination of large RoM in the knees and lower muscle power may make children who are underweight more prone to injuries. 

### 6.6. Hypermobility Spectrum Disorder

Clinically, a higher frequency of pain exists in hypermobile children compared to normal mobile [2], however, this was not found in our sample. None of the children were classified as having Hypermobility Spectrum Disorder according to our assessments. Hypermobile children did not report pain; in fact, hardly any pain was reported in this random population. 

### 6.7. Strengths and Limitations of the Study

The inclusion of goniometric RoM of the joints in this study provides additional value to the Beighton criteria in identifying joint hypermobility. This may help identify individuals within the upper limit of joint hypermobility, thus improving both the sensitivity and specificity of the Beighton scoring system. It may be a more sensitive way of determining the severity of joint hypermobility other than the previously used dichotomous scale. In addition, it may also provide insight into whether the various excess degrees of movement increase the risk of developing musculoskeletal symptoms in a group of individuals with GJH. 

Our study included a large number of Nigerian children, however, it is important to replicate the study in other African and Western countries, to be able to generalize the findings.

### 6.8. Future Studies

Previous studies have tried to establish a causal relationship between GJH and the onset of musculoskeletal symptoms without consideration for the largeness of excess degree of motion; this needs to be pursued in new studies. Moreover, the question remains to be answered if individuals, who are classified as normal mobile but with localized hypermobile joints, will develop symptoms.

## 7. Conclusions

This was the first study to describe the prevalence of all three classifications of asymptomatic joint hypermobility (PJH, LJH, GJH) and HSD in school-aged children. Although a prevalence of 64.7% hypermobility is high (cut-off ≥5), we did not find any children with Hypermobility Spectrum Disorder. Whether this will change when children grow older will be the topic of our longitudinal study. Our findings extend previous observations that the item hand-on-floor gives divergent results; it is the only item with a low frequency in hypermobile children, it is not sensitive to age, and it showed a trend to be more frequent in boys than girls.

Taken together, we concluded that in this age bracket, hypermobility is more the rule than the exception. It is often based on excessive RoM in the finger joints (58%). Additionally, mobility changes with age. Differences between genders in the RoM in our sample follow the described patterns in the literature. Goniometry, instead of the dichotomous rating of the joints, will provide more sensitive information. Clinically, this may be important in the study of the development of musculoskeletal complaints later in life. The criteria for defining GJH in any given population will determine its prevalence. Consensus on Beighton score cut-off values as appropriate for age, gender, and ethnicity is urgently needed. A minimum requirement for all studies reporting on hypermobility is that the cut-off values are mentioned. 

## Figures and Tables

**Figure 1 healthcare-11-01092-f001:**
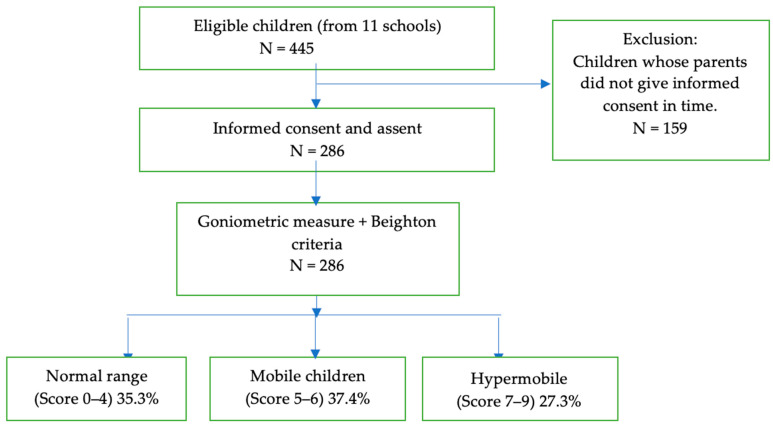
Participant recruitment flowchart depicting the steps in the selection procedure of the children and results per the classification of joint mobility.

**Figure 2 healthcare-11-01092-f002:**
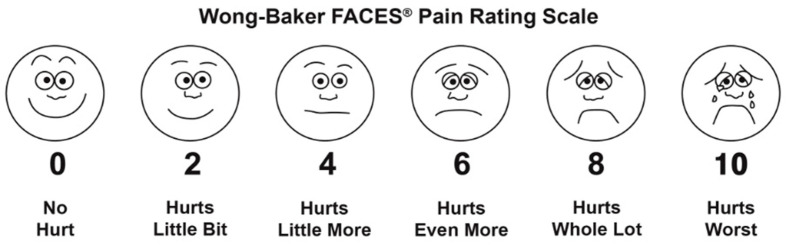
Wong–Baker Faces Pain Rating Scale. Used with permission, originally published in Whaley & Wong’s Nursing Care of infants and children.

**Figure 3 healthcare-11-01092-f003:**
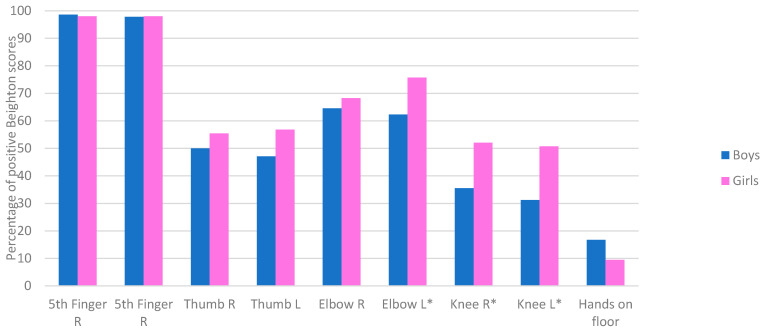
Percentage of positive Beighton scores for boys and girls per item. * indicates significant differences.

**Figure 4 healthcare-11-01092-f004:**
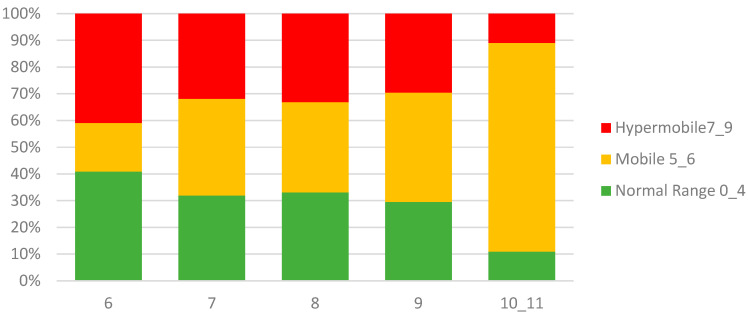
Percentage per Beighton classification for each age group.

**Figure 5 healthcare-11-01092-f005:**
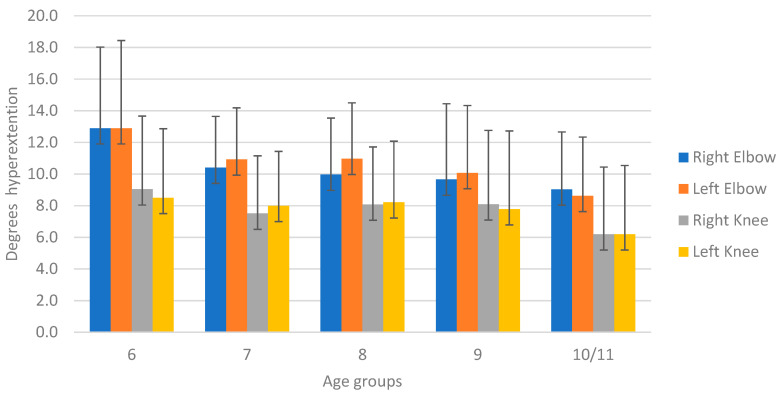
Range of motion in the age groups based on goniometry.

**Figure 6 healthcare-11-01092-f006:**
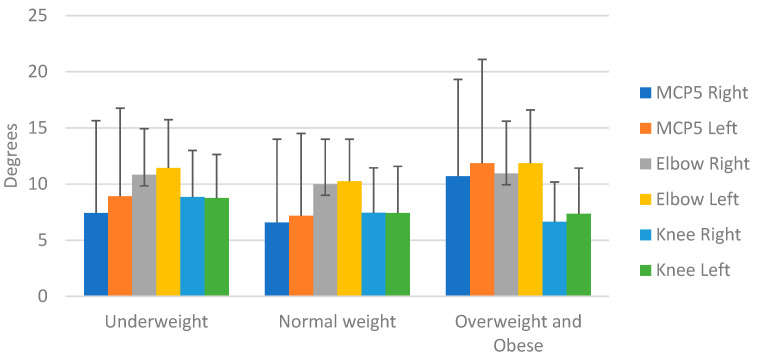
Range of motion per joint for the BMI classification groups.

**Figure 7 healthcare-11-01092-f007:**
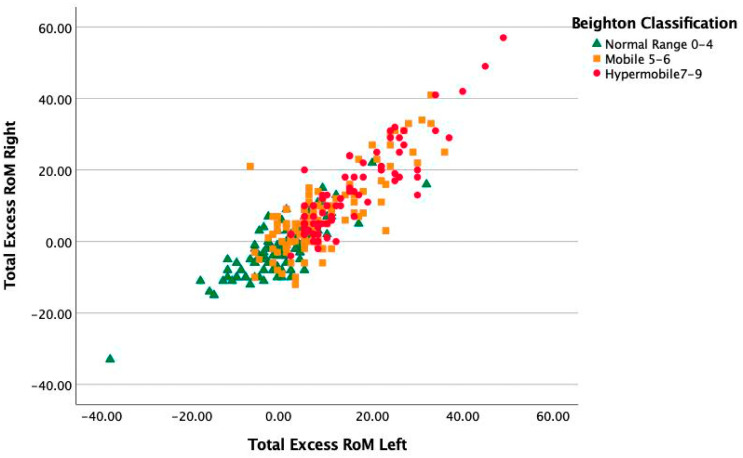
Excess range of motion. Note, there is a large overlap in the excess range of motion in the three Beighton classification groups.

**Table 1 healthcare-11-01092-t001:** Beighton scoring system.

Items	Right	Left
a. Passive opposition of the thumb to the volar side of the forearm	1	1
b. Passive dorsiflexion of the 5th MCP joint to ≥90^0^	1	1
c. Passive hyperextension of the elbow joint to ≥10^0^	1	1
d. Passive hyperextension of the knee joint to ≥10^0^	1	1
e. Placing hands flat on the floor with the knees kept straight	1	
Maximum possible score	9	

**Table 2 healthcare-11-01092-t002:** Demographic distribution of the joint mobility groups (Beighton score range).

Demographics	Normo-Mobile (Beighton 0–4) N (%)	Mobile (Beighton 5–6)	Hypermobile (Beighton 7–9) N (%)	Test Value,
		N (%)		*p*-Value
Age years				Χ² 27.34, *p* 0.007
6	11 (22.9)	14 (29.2)	13 (47.9)	
7	28 (35.4)	31 (39.2)	20 (25.3)	
8	28 (31.8)	35 (39.8)	25 (28.4)	
9	15 (36.6)	17 (41.5)	9 (22)	
11–10	19 (63.3)	10 (33.3)	1 (3.3)	
Mean (age)	8.03 (1.28)	7.79 (1.15)	7.29 (1.06)	F 8.96, *p* 0.001
Gender				Χ² 7.55, 0.023
Male	57 (41.3)	53 (38.4)	28 (20.3)	
Female	44 (29.7)	54 (36.5)	50 (33.8)	
BMI class				Χ² 3.45, 0.485
Underweight	35 (31.5)	40 (36)	36 (32.4)	
Normal weight	53 (37.6)	56 (39.7)	32 (22.7)	
Overweight	13 (38.2)	11 (32.4)	10 (29.4)	
Mean BMI	15.86 (3.35)	15.16 (2.79)	14.93 (2.93)	F 2.36, 0.097

N = Number of children per classification; (%) percentage of children per classification; BMI class = BMI classification. Alpha was set at 0.05.

**Table 3 healthcare-11-01092-t003:** Excess degrees of movement in the 5th fingers, elbows, and knees in children classified as normal RoM, mobile and hypermobile.

Beighton Classification	Excess RoM Right Mean (Range) in Degrees	Excess RoM Left Mean (Range) in Degrees
Normal Range (n = 101)	−2.3 (−33–22)	−0.78 (−38–32)
Mobile (n = 107)	7.3 (−12–41)	8.9 (−7–36)
Hyper Mobile (n = 78)	14.0 (−4–57)	15.4 (2–49)

## Data Availability

The datasets used and/or analyzed during the current study are available from the corresponding author upon reasonable request.

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
