# Peer review of "Prevalence and Demographic Distribution of Hypermobility in a Random Group of School-Aged Children in Nigeria"

_healthcare, 2023, doi:10.3390/healthcare11081092_

Round 1
Reviewer 1 Report
Minor comments
1. The conceptual framework (Figure 1) should be in one page only
2. A few sentences should be inserted to describe the Figure 1 in the section 'Participant Description'
3. Please, insert the values for each BMI as recommended by the WHO). Also cite the WHO References for the BMI, as you you know there are changes, as different countries suggest their personal demographics for the domestication of BMI...(Check China have its own BMI values, as against the WHO standard and recommended BMI) [The body mass 172 index (BMI) calculation was performed using a metric formula, weight (in kilograms) di-173 vided by height (in meters squared)].
4. Get the child's BMI and not the Adult's BMI....differentiate them? and if they are the same, also state it in your methods
5. How was the 9 points Beighton score divided into 3 main categories? Can you briefly explain it for clarification to your reader of interest?
6. Check line 195 ...what does that figure mean in the citing reference?
7. Please, can you briefly define or explain the following Normo- mobile, Mobile, and Hypermobile?
8. State the strengths and limitations of the study
9. If possible, you can send the paper to a professional english language editor for editing
10, Mind font spacing in between sentences
Reviewer 2 Report
Dear Editor,
Thank you for the opportunity to review the article Prevalence and demographic distribution of hypermobility in a randomized group of 2 school children in Nigeria whose objective was to determine the prevalence of hypermobility 10 in randomly selected healthy children without previous trauma or disease process affecting the 11 joints and whether other demographic variables.
I would like to congratulate the authors for the quality of the article, namely the high substantiation of the theme in the introduction.
The objectivity and systematisation of the contents in the methods.
Justification only in the measuring instruments, if they are validated for the population in question.
I would suggest greater investment in the graphs and think about the need for so many figures.
In relation to the discussion, within the scope of the Beighton scoring system, I would suggest greater discussion in relation to the finding being superior in girls.
In the discussion, at the end, explore the implications of the study for practice and for the clinic, as well as adding its respective limitations.
